# Biomolecular Actions by Intestinal Endotoxemia in Metabolic Syndrome

**DOI:** 10.3390/ijms25052841

**Published:** 2024-02-29

**Authors:** Ioannis Alexandros Charitos, Maria Aliani, Pasquale Tondo, Maria Venneri, Giorgio Castellana, Giulia Scioscia, Francesca Castellaneta, Donato Lacedonia, Mauro Carone

**Affiliations:** 1Istituti Clinici Scientifici Maugeri IRCCS, Pneumology and Respiratory Rehabilitation Unit, “Istitute” of Bari, 70124 Bari, Italy; maria.aliani@icsmaugeri.it (M.A.); giorgio.castellana@icsmaugeri.it (G.C.); mauro.carone@icsmaugeri.it (M.C.); 2Department of Medical and Surgical Sciences, University of Foggia, 71122 Foggia, Italy; pasquale.tondo@unifg.it (P.T.); giulia.scioscia@unifg.it (G.S.); donato.lacedonia@unifg.it (D.L.); 3Institute of Respiratory Diseases, Policlinico Riuniti of Foggia, 71122 Foggia, Italy; 4Istituti Clinici Scientifici Maugeri IRCCS, Genomics and Proteomics Laboratory, “Istitute” of Bari, 70124 Bari, Italy; 5School of Clinical Biochemistry and Pathology, University of Bari (Aldo Moro), 70124 Bari, Italy; francesca.castellaneta@gmail.com

**Keywords:** metabolic syndrome (MetS), biochemistry, molecular biology, human microbiota, metabolome, immunity, microbiota’s crosstalk axis, chronic obstructive pulmonary disease (COPD), respiratory rehabilitation, probiotics, prebiotics

## Abstract

Metabolic syndrome (MetS) is a combination of metabolic disorders that concurrently act as factors promoting systemic pathologies such as atherosclerosis or diabetes mellitus. It is now believed to encompass six main interacting conditions: visceral fat, imbalance of lipids (dyslipidemia), hypertension, insulin resistance (with or without impairing both glucose tolerance and fasting blood sugar), and inflammation. In the last 10 years, there has been a progressive interest through scientific research investigations conducted in the field of metabolomics, confirming a trend to evaluate the role of the metabolome, particularly the intestinal one. The intestinal microbiota (IM) is crucial due to the diversity of microorganisms and their abundance. Consequently, IM dysbiosis and its derivate toxic metabolites have been correlated with MetS. By intervening in these two factors (dysbiosis and consequently the metabolome), we can potentially prevent or slow down the clinical effects of the MetS process. This, in turn, may mitigate dysregulations of intestinal microbiota axes, such as the lung axis, thereby potentially alleviating the negative impact on respiratory pathology, such as the chronic obstructive pulmonary disease. However, the biomolecular mechanisms through which the IM influences the host’s metabolism via a dysbiosis metabolome in both normal and pathological conditions are still unclear. In this study, we seek to provide a description of the knowledge to date of the IM and its metabolome and the factors that influence it. Furthermore, we analyze the interactions between the functions of the IM and the pathophysiology of major metabolic diseases via local and systemic metabolome’s relate endotoxemia.

## 1. Introduction

### 1.1. The Metabolic Syndrome (MetS)

Metabolic syndrome (MetS) comprises a group of related diseases, including obesity, cardiovascular disease, non-alcoholic fatty liver disease (NAFLD), and type II diabetes. Comorbidities and common risk factors for MetS features include a proinflammatory state, prothrombotic state, steatosis, reproductive disorders, excessive abdominal fat, elevated triglyceride levels, low HDL, arterial hypertension, insulin resistance, and intolerance glycaemia [1]. Each of these risk factors is considered a risk factor for the other pathologies or disorders, such as cardiac ischemia, obstructive sleep apnea syndrome, dementia, etc. The global prevalence of MetS is increasing and is estimated to range from approximately 12.5% to 31.4% in the adult population worldwide, highlighting the need to control the risk factors for the development and progression of diseases associated with MetS [2]. The intestinal microbiota (IM) is characterized by great inter-individual variability in composition, function, and bio humoral interaction with the host, making it a “singular personal track”. This characteristic contributes to individual susceptibility to the development of distinct complications of MetS. The composition and function of the IM are considered important factors with positive associations for the glucose homeostasis. For instance, studies noted that subjects with high levels of the *Prevotella* genus had better blood glycemia control [3,4]. A functional examination of the IM revealed that *Prevotella* appears to contribute to positive outcomes by aiding in the breakdown of dietary-fiber-related enzymes found in bread and enhancing glycogen storage. On the other hand, certain combinations of bacteria may have individualized negative effects on their host in response to dietary personal attituding [5]. The consumption of calorie-free artificial sweeteners, which could be lead to IM dysbiosis, has been linked to promoting MetS in vivo (in mice) studies. The ability of calorie-free artificial sweeteners to promote glucose intolerance is influenced by the host IM. Despite numerous associations linking the IM to overall metabolic responses and the various morbidities associated with MetS, the underlying biochemical mechanisms are highly intricate [6]. Evidence indicates that inflammation and certain metabolites found in the IM, including short-chain fatty acids and bile acids, play a substantial role in impacting the advancement of disorders and diseases associated with MetS [7,8].

Differences between individuals in the risk of developing MetS, response to diet, and medical treatment are often attributed to hereditary (genetic) factors, lifestyle, and age [9]. Therefore, the makeup of microorganisms within the IM and their interaction with the host can impact diverse physiological functions, playing a pivotal role in the susceptibility to numerous diseases and disorders, including MetS. The IM is the total community of microorganisms, and is a part of the gastrointestinal tract microbiota, consisting of bacteria, viruses, fungi, and protozoa that coexist in a “harmonious relationship”, leading to a condition of stability called eubiosis (Greek = εὐ (well) + βίωσις (to live) = well-being), providing benefits to the organism (Figure 1) [9,10].

If this condition of coexistence, for various reasons, is disturbed (disproportionate), there will be a symbiotic disadvantage known as dysbiosis (Greek = δὐς (difficulty) + βίωσις (to live) = not harmonious coexistence) among the microorganisms. Hence, alterations in the population levels of a particular bacterial genus are likely to influence the populations of other genera, disrupting the overall state of eubiosis and thereby causing qualitative and quantitative changes in the metabolome [11]. This can lead to an increase in derivate-IM toxic metabolites. Indeed, the correlation between the dysbiosis of the ΙΜ and the appearance of metabolic disorders is known to increase the possibility of a dysbiosis metabolome that are not related to the health of the host. What will happen will be an accumulation of metabolic toxic products by bacteria at the intestinal level (local endotoxemia) which creates local chronic inflammation [12]. In fact, we have the presence of certain cytokines such as Interleukin (IL)-22 (belongs to IL-10 expressed predominantly by innate lymphoid cells (ILC) and T helper (TH) CD4+) acts to trigger antimicrobial immunity and maintain the integrity of the mucosal barrier within the intestine. Finally, this pathological condition in turn will lead to a series of events, such as defects in the preservation of the integrity of the mucosal barrier which can cause systemic endotoxemia to the downregulation of the microbiota’s intestine-system axes and further promotes the development of MetS (Figure 2) [13,14,15].

Research conducted by Jeffrey Gordon’s team revealed variations in the IM between obese and lean subjects, highlighting its substantial impact on the development of obesity through its interaction with the host. Subsequent investigations have further elucidated the association between IM dysbiosis and other MetS-related conditions such as type 2 diabetes, NAFLD, and atherosclerosis.

Furthermore, the IM enhances the host’s resistance to colonization by exogenous pathogens or the development of opportunistic ones. Intestinal microorganisms work competitively against these bacteria by preventing their invasion and keeping their populations low. This competitive advantage stems from (a) their competition with pathogens for available energy sources, (b) they produce toxic antibacterial substances and H_2_O_2_, and (c) the secretion of extracellular enzymes that inhibit the attachment of pathogens to their receptors [16,17]. Therefore, the IM assumes a significant role in fostering the development and optimal operation of the human immune system, encompassing both innate and acquired immunity. Via the enteric nervous system (ENS), it engages with the immune–metabolic system axis and the assortment of regulatory mechanisms posited to elucidate its impact on health and susceptibility to (Figure 3). A key player in this condition is the gastrointestinal-associated lymphoid tissue (GALT), which is the lymphoid tissue associated with the intestine, normally part of the immune system present in the gastrointestinal tract. The GALT encompasses a part of the mucosal-associated lymphoid tissue (MALT), whose function is triggered by components of the gastrointestinal microbiota via antigenic stimuli, thereby influencing immune biochemical processes [18,19]. Numerous B and T lymphocytes are typically found in the intestinal tract. This lymphoid tissue is responsible for protecting mucous membranes from attacks by pathogens, which could lead to infections, sepsis, and other conditions, both in the primary and secondary response. Indeed, *Fusobacteria* genes have been identified as proinflammatory, activating bone marrow-derived immune cells (such as macrophages, dendritic cells, and suppressor cells), histone apo-acetylases and butyrate appears to suppress the nuclear factor κB (NF-κB) [20,21].

As mentioned earlier, another factor predisposing to MetS is genetics. Genetic factors play a role in the development of MetS. Among the species associated with the MetS condition is the phylum *Actinomycetota* (to which *Bifidobacterium* belongs). Reduced concentrations of *Actinomycetota* were significantly associated with the presence of the minor allele in the APOA5 rs651821 SNP [22]. Single nucleotide polymorphisms (SNPs) have been identified in the genes encoding lipoprotein lipase, cholesterol transfer protein, and apolipoprotein A (APOA). The APOA5 gene, found in proximity to the APOA1/C3/A4 gene cluster on chromosome 11, plays a role in regulating triglyceride metabolism. It codes for apolipoprotein A-V, which is a constituent of chylomicrons, very low-density lipoprotein, and high-density lipoprotein (HDL) particles [23]. SNPs in the APOA5 gene are associated with elevated plasma triglyceride levels. Specifically, the SNP rs651821 (located 3 bp upstream of the APOA5 start codon) has shown significant associations with IM and the risk of hypertriglyceridemia and MetS. Each additional copy of the C allele on the APOA5 SNP rs651821 was noted to reduce the concentrations of the *Bifidobacteriaceae*. However, the SNP rs651821 polymorphism of the APOA5 gene may be an important genetic factor in determining the health-promoting Bifidobacterium bacteria [24]. Regarding Type 2 diabetes mellitus, the *Bacteroides coprocola* strain was found to be abundant. The genes EDU99824.1 and EDV02301.1 encode glycosylation hydrolases, which contribute to the degradation of cellulose and starch, resulting in the creation of sugars. It is worth noting that alpha-glucosidase, a glycosyl hydrolase positioned at the small intestine’s edge, stands out as a significant pharmaceutical target for managing type II diabetes [25]. SNPs in the INS and IFIH1 genes have been associated with Type 1 diabetes. Genome-based approaches for identifying genetic polymorphisms associated with Type 1 diabetes (such as IL-2RA and CUX2) have revealed specific SNPs (such as PTPN22, IL-10, IL-27, and IL18RAP) with contrasting effects on the development of Type 1 diabetes [26]. This suggests that the regulation of a particular pathological immune pathway can pose a risk for one disease while being protective for another. For instance, the PTPN22 (R602W) variant increases the susceptibility to Type 1 diabetes. Conversely, the SNP rs3024505 within IL-10 and the rs4788084 locus near IL-27 and NURP1 genes exhibit protective roles against the onset of Type 1 diabetes [27].

### 1.2. Metabolomics Concepts

Beyond describing bacterial community composition and its associated disease, research on the IM is progressing toward elucidating activated molecular pathways and metabolome, as well as characterizing their effects on MetS-related events in the host. These investigations involve a blend of next-generation sequencing, metabolomics methodologies, and in vivo experiments using mouse models. Their collective goal is to pinpoint bacterial communities, variations in microbial species levels, host genes, transcripts, and metabolites alterations [28]. These examinations can contribute to the advancement of more precise diagnostic and treatment strategies for pathologies related to Metabolic Syndrome in the future. Genomics encompasses the thorough examination of DNA structure and function. Grasping biodiversity at the genome-wide scale will enhance our comprehension of the origins of individual traits and susceptibility to diseases [29]. It involves the examination of DNA polymorphisms and mutations, as well as the global analysis of gene expression through the sub-discipline of transcriptomics. Proteomics, on the other hand, entails the systematic study of proteins to offer a thorough understanding of a biological systems’ structure, function, and regulation [30]. Metabolomics is the systematic study of the metabolome, representing the set of all metabolites involved as intermediates and/or final products of the biochemical processes of a cell, tissue, organ, or organism. Data on gene expression and proteomic analysis, by themselves, cannot provide a comprehensive description of the ongoing processes, whether of endogenous origin or induced by an exogenous stimulus [31]. The metabolic profile, in contrast, can be seen as the final product of gene expression and protein (enzyme) activity, thus defining the biochemical phenotype of a biological system. While genomics and proteomics suggest a possible way of functioning of a biological system, metabolomics provides a current representation that is closer to reality [32]. Genes alone do not predict the phenotype, which is the result of the genetic constitution, environmental influences, and pathophysiological conditions that can overlap during the life cycle of an organism. Hence, while genomics, transcriptomics, and proteomics provide insights into potential causes of a phenotypic response, they do not forecast the subsequent level of occurrence [33]. Metabolomics, as the study of the profile of the metabolites of a biological system, provides a functional vision of an organism. This results from the genetic structure, gene expression through RNA production, protein activity, and interaction with environmental factors (e.g., lifestyle, diet, pharmacological treatments, abuse of substances, etc.) [34,35]. It can be considered a form of functional genomics in which metabolites constitute the basis on which all cellular processes are built, reflecting the state of functional equilibrium of biological events in each system. Metabolomics is characterized as the “quantitative assessment of the multiparametric metabolic-dynamic reaction of living systems to pathophysiological stimuli or genetic alteration”. Meanwhile, it delves into the examination of the “entire array of low molecular weight metabolites/intermediates, reflecting the physiological dependencies, developmental stages, and pathological conditions of the cell, tissue, organ, or organism. Currently, these two terms are utilized interchangeably and have become synonymous [35]. As mentioned above, metabolomics can provide a profile of the metabolic state, offering a global understanding of cells, tissues, and organisms within both physiological and pathological conditions [36,37]. All biological processes, whether natural or induced by ischemia, drugs, or hormones occur through a sequence of biochemical reactions/interactions that generate a vast and complex set of metabolites. A considerable number of these metabolites are released into the bloodstream and urine, where they can be detected using specific methods. The techniques used in metabolomics can detect relatively low molecular weight molecules (<1000 Da), including nucleotides, oligopeptides, amino acids, fatty acids, lipids, carbohydrates, vitamins, and organic acids. These analyses are carried out on biofluids (plasma, urine, cerebrospinal fluid, saliva, etc.) and on intact tissues (heart, liver, kidney, brain, etc.) [38,39]. The metabolomic approach consists of two sequential phases: analysis methods and data processing and interpretation. The primary experimental methods employed in metabolomics include nuclear magnetic resonance (NMR) spectroscopy and mass spectrometry (MS) [40,41,42].

## 2. The Importance of IM Composition over Time

There is conflicting scientific evidence regarding issues related to the fetal/placental microbiota. Some evidence supports the “sterile womb” hypothesis. Recent studies have indicated that fetal intestinal microorganisms may be affected during pregnancy due to the presence of bacteria in the amniotic fluid, placenta, and umbilical cord (Figure 4) [43,44].

Several authors, including Blaser MJ and colleagues, argue that the detection of low levels of bacterial DNA in clinical samples may be attributed to contamination of PCR, tissues, or reagents, or to the actual presence of bacterial DNA in the subjects’ blood and its subsequent amplification. The current scientific evidence does not strongly support true microorganisms’ colonization in utero and beyond, nor does it support the concept of a genuine “microbiota” in the placenta or amniotic sac, which can be considered a community of microorganism and a set of interacting, often interdependent species [44,45]. Various factors have been studied that can influence the development of the fetal IM during pregnancy, such as maternal stress, antibiotic intake, and birth pattern. The mode of childbirth, whether natural or by caesarean section, has been shown to influence the initial exposure of the newborn to microorganisms’ colonization, resulting in different microbiota compositions. However, some metabolites (such as SCFAs) have been implicated in metabolic disorders during pregnancy which appear to play a fundamental role in maternal health and neonatal development (Figure 4) [46,47,48].

In childhood, new bacterial colonize the gut while other are replaced. This period is marked by instability in the maturation process of the IM, influenced by dietary habits and diseases. In adulthood, the IM stabilizes in composition and varies individually. In old age, it undergoes further changes, distinctly differentiating from the microbiota of younger individuals [49]. The development of a child’s microbiota into a relatively stable composition can be divided into four characteristic stages, beginning from the first year, when breast milk intake plays a crucial role in supporting the immune system’s development. During adulthood, up to 50% of the colon’s microbial volume comprises archaea and bacteria, mainly anaerobic. The main anaerobic phyla are *Bacteroidota*, *Actinomycetota* (such as *Bifidobacterium* ssp., *Propionibacterium* spp.), *Bacillota* (such as *Eubacterium* spp., *Clostridium* spp.), while facultative aerobic ones mainly belong to *Pseudomonadota* (such as *Escherichia coli*) [49,50].

Several host factors, such as comorbidities, superinfections, malnutrition, and drugs such as antibiotics and antivirals, can lead to further microbiota dysbiosis and alterations of metabolome [7,32]. The ongoing SARS-CoV-2 pandemic has been shown to induce dysbiosis, particularly in the gastrointestinal and lung microbiota [51]. Approximately 2–10% of patients have gastrointestinal symptoms (such as abdominal pain, nausea, vomiting, diarrhea), respiratory and skin disorders (such as urticaria), alongside neurological (such as intracerebral bleeding) and psychiatric disorders (such as psychosis and affective disorders) [52]. Remarkably, there is a substantial proliferation of opportunistic pathogenic populations within the IM, including *Streptococcus* spp., *Veillonella* spp., *Clostridium hathewayi*, *Clostridium ramosum*, *Actinomyces viscosus*, and *Bacteroides nordii*. In the context of the SARS-Cov-2 worldwide infection, this will lead to the hypothesis of the “dysbiosis cycle of immunity dysregulation” (IDDC). Consequently, patients may exhibit dysregulation in bidirectional link axes, such as the intestinal/lung, gut/brain, gut/skin, and other. As a result, immune dysregulation increases, leading to continuous local dysbiosis in the IM and immune system dysregulation, likely worsening the clinical outcome for the patient [52,53].

Finally, the intestinal environment creates several differences in microorganism communities. Condition change in each area of the gastrointestinal tract, affecting their survival. In the large intestine, communities demonstrate better stability than in the ileum. Intercellular variability between microorganism communities is influenced by internal (genetic factors, age, gender, exposure to stress, health status, etc.) or external factors. Physical and/or mental stress is a significant factor causing changes in the IM and in the metabolome [54]. Additionally, differences in the metabolome composition of the IM can be observed between individuals living in different parts of the world. The main parameters affecting the ecological sites, survival, and activity of bacteria strains include pH, temperature, local permeance time, peristalsis, presence of bioactive lumen and cellular enzymes, glycoproteins, lipids, amino acids, and carbohydrates (except for pH), the presence of other local strains, and the mucus viscosity [55]. In adults, the colon’s microbiota is more complex than that of the children. As age increases, the population of *Bifidobacteria* decreases, ranking third in abundance after the genera *Eubacterium* and *Bacteroides.* These differences are also evident in the composition of feces in adults, characterized by a low redox potential (Eh), a neutral or slightly alkaline pH, and the presence of significant amounts of degradation products such as ammonia, amines, phenols, and degraded bile acids [56].

## 3. The IM-Derived Metabolites

The metabolome, produced, degraded, or modulated by the IM, serves as a “communicator provider” between the host and the microorganism’s community. Short-chain fatty acids, bile acids, and trimethylamines are among the extensively studied metabolites, while there exist numerous other metabolites yet to be explored, which could profoundly influence the host’s metabolism (Figure 5) [57]. Obesity, insulin resistance, atherosclerosis, and steatohepatitis are disorders associated with MetS and inflammation. Adipose tissue inflammation is extensively researched concerning obesity and Type 2 diabetes, playing a role in the diseases’ pathogenesis through both innate and adaptive immune responses [58]. Bacteria endotoxins, such as lipopolysaccharides (LPS), have been detected in Type 2 diabetes, obese, and insulin-resistant mice, leading to increased adipose and systemic inflammation [59,60]. Hence, dietary choices significantly influence how the gut microbiome impacts inflammation linked to MetS. Mice consuming lard trigger lipid peroxidation receptor (TLR) signaling and inflammation, leading to elevated serum LPS levels and adiposity. This metabolic impact can be transmitted to germ-free (GF) mice. The metabolic phenotype of mice fed fish oil rather than lard was compensated, indicating that diet plays an important role in microbial composition, subsequently modulating inflammation, and adiposity in adipose tissue [61,62]. A key link between intestinal inflammation, intestinal microorganism’s alterations, and Nonalcoholic Fatty Liver Disease has also been suggested [63]. In an investigation, mice lacking inflammasome signaling showed alterations in IM composition linked to hepatic steatosis. This was instigated by a substantial influx of TLR4 and TLR9 agonists, resulting in heightened hepatic TNFα secretion, liver injury, and inflammation. Metabolic outcomes were synergistically transferable, suggesting a significant overlap between IM and hosts in the evolution of NAFLD. Another research finding indicates that the bile acid taurine regulates microbiome composition, thereby triggering NLRP6 inflammasome activation. Mice administered with taurine experienced improvements in colitis symptoms, a change contingent upon the intestinal microbiota (IM) and inflammasome activation. However, the impact of taurine on metabolic complications is yet to be fully elucidated [16,64,65,66,67,68,69].

Trimethylamine (TMA) emerges as a byproduct of L-carnitine metabolism, abundant in red meat, and phosphatidylcholine conversion, prevalent in cheese and eggs. TMA journeys to the liver through portal circulation, where flavone monooxygenases (FMO) convert it into TMA N-oxide (TMAO), a pro-atherogenic compound linked to coronary artery disease and thrombosis in both mice and humans. Mice treated with antibiotics or GF mice exhibited untraceable levels of TMA and TMAO, while conventional mice displayed heightened TMAO levels, highlighting the indispensable role of IM in TMAO synthesis [70]. Consequently, antibiotic-treated or GF mice fed L-carnitine or phosphatidylcholine diets manifested reduced atherosclerotic lesions, diminished foam cell accumulation, and decreased platelet hyperactivity. The IM’s pivotal role in TMAO generation was further validated in humans treated with L-carnitine or phosphatidylcholine and antibiotics, resulting in near-total suppression of TMAO. The potential direct clinical implications of TMAO in cardiovascular disease development and progression necessitate further prospective investigations [71].

Short-chain fatty acids (SCFAs), such as acetate, propionate, and butyrate, arise from the bacterial breakdown of polysaccharides within the gastrointestinal tract through fermentation processes. SCFAs may play a role in maintaining body weight, intestinal homeostasis, and improving lipid and glucose metabolism. In most in vivo (animal) studies, dietary supplementation with SCFAs reduced weight gain and triglycerides and improved insulin sensitivity (Figure 6) [72]. 

The administration of propionic acid to overweight, abdominal fat, fatty liver, and insulin resistance, while significantly increasing postprandial secretion of Peptide YY or PRR (a hormone produced in the L cells of the intestinal mucosa of the ileum and colon) and Postprandial glucagon-like peptide-1 (GLP1) [66,73]. SCFAs serve as ligands for G-protein-coupled receptors GPR41, GPR43, and GPR109a, which are present in the colonic epithelium, pancreatic β-cells, and adipose tissue. Mice deficient in GPR41 exhibited leanness and reduced expression of the gastric incretin PYY. Acetate and propionate act as potent ligands for GPR43, and GPR43-deficient mice on a high-fat diet (HFD) gained more weight with increased complications associated with MetS [74]. Antibiotic treatment under germ-free (GF) conditions abolishes the metabolic effects observed in GPR41 and GPR43 mice, suggesting that bacterial SCFAs induce the activation of GPR43 and GPR41, which regulate whole-body energy and glucose homeostasis. Additionally, the host response to SCFAs may involve glucose sensing through the intestinal-brain axis and immunomodulation, such as the intestinal-lung axis (Figure 7) [75].

This relationship hypothesis interaction between MetS and the intestinal-lung axis can be demonstrated by some studies which observed that pathogenic metabolites can influence the course of respiratory diseases such as Chronic obstructive pulmonary disease (COPD). Indeed, in a study with sixteen males affected by COPD, the metabolomic, oxidative, and inflammatory responses to constant and intermittent work rate exercises in plasma were studied. Individuals performed symptom-limited incremental cycle exercise testing (ICE). Acute CE and IE alter circulating GPx levels in COPD, indicating the shift of energy metabolism from carbohydrates to amino acid utilization and lipid metabolism in COPD. From this, it can be deduced that the derivate metabolites of the IM can have both positive and negative influences during pulmonary or cardiac rehabilitation in patients with MTS (Table 1) [76].

Propionic acid sensing in the colon induces intestinal gluconeogenesis, leading to improved glucose and weight regulation. HFD rats had increased colonic acetic acid incorporation, and chronic acetic acid administration induced obesity-related MetS complications and led to parasympathetic β-cell stimulatory insulin secretion [66,77,78]. Microbial metabolic byproducts like SCFAs interact with G-protein-coupled receptors (GPCRs) on intestinal epithelial cells, including Gpr41 and Gpr43, to modulate energy balance. This interaction involves the intestinal hormone Pyy and influences the host’s inflammatory response. Activation of Tlr5, possibly by bacterial flagellum, impacts the structural makeup of the IM, thereby influencing appetite, weight gain, and insulin sensitivity through mechanisms that are not yet fully understood. Nonetheless, the role of SCFAs holds promise for treating metabolic syndrome [79,80].

Bile acids are predominantly synthesized through hepatic cholesterol breakdown, then conveyed to the gallbladder and intestinal lumen through peristalsis. Within the distal small intestine and colon, the intestinal microbiota (IM) can convert primary bile acids into secondary bile acids [81,82]. The physiological effects of bile acids are largely mediated by the G protein-coupled receptor TGR5 and the farnesoid nuclear receptor (FXR). FXR, a transcription factor, regulates endogenous bile acid synthesis and release, with its activation inhibiting hepatic bile acid biosynthesis [82,83]. In obese and insulin-resistant mice, reduced IM diversity correlates with diminished secondary bile acids and hepatic enzymes involved in bile acid biosynthesis, alongside increased FXR and decreased TGR5 expression. TGR5 activation by bile acids enhances insulin sensitivity, while FXR binding to bile acids lowers liver cholesterol levels and serum triglycerides [84,85]. Studies involving germ-free (GF) and antibiotic-treated mice suggest that the metabolome can modulate FXR and related genes controlling bile acid synthesis. Intestinal FXR-deficient mice on a high-fat diet (HFD) display decreased body weight, glucose intolerance, and insulin resistance, along with protection against hepatic steatosis development. Blockade of intestinal FXR by administration of an FXR antagonist promoted gray fat, reduced adiposity, and insulin resistance. All these animal studies indicate a dominant role for the IM in regulating bile acids, polymorphism, and FXR signaling and modulating the complications of MetS [86,87].

## 4. Metabolome’s Pathogenic Biomolecular Mechanics in MetS

### 4.1. Obesity

Obesity triggers immune dysfunction characterized by the release of inflammatory adipokines such as TNF-α, IL6, and leptin from adipose tissues [88]. These inflammatory adipokines can instigate carcinogenic mechanisms like enhanced cell proliferation or dedifferentiation, posing potential risks for cancers like colon, esophageal, and hepatocellular carcinoma. Intra-abdominal adipose tissue secretes adipokines with atherogenic properties (IL-1, IL-6, TNF-α, and IFNα), heightening the susceptibility to cardiovascular diseases [88,89]. These pro-inflammatory cytokines also activate specific kinases, inducing the expression of inflammatory and adipogenic genes, thereby amplifying inflammation and increasing adipose tissue mass (increased adipocyte size) [88]. Reduced expression of peptides like GLP-1 and GLP-2, which regulate mucosal barrier function, results in altered mucosal function and diminished synthesis of tight-binding proteins Zonula Occludin-1 and -2 (ZO-1, ZO2), leading to increased intestinal permeability [90,91]. This heightened permeability allows LPS to enter systemic circulation, triggering the release of pro-inflammatory cytokines. Consequently, proinflammatory cytokines activate kinases like JNK and IKK, boosting the expression of inflammatory cytokines and lipids. To investigate the hypothesis that host genetics can influence microbial adiposity, candidate genes have been analyzed, and single nucleotide polymorphisms (SNPs) associated with obesity have been selected [92]. The strongest association between host genotype and Adiposity Measures and Operational Taxonomic unit (OTUs) was observed between Greengenes out 181702 within the order *Clostridiales* (belong the families of *Acidaminococcaceae*, *Clostridiaceae*, *Eubacteriaceae*, *Heliobacteriaceae*, *Lachnospiraceae*, *Peptococcaceae*, *Peptostreptococcaceae,* and *Syntrophomonadaceae*), and a host genetic variant within an intron of the Fragile Histidine Triad Diadenosine Triphosphatase (FHIT) gene (rs74331972) [93,94]. FHIT encodes the protein histidine, and has been linked to cancers of the digestive system. The most significant FHIT association was obtained with OTU 181702, which was significantly associated with SFM (subcutaneous fat mass) and VFM (visceral fat mass). The second significant genetic association between variants was near the Testis-development-related protein 1 (TDRG1) gene with *Clostridiales* OTU, which was identified as significantly associated with VFM [95]. The final significant genetic association was observed in a variant in an intron of the ELAVL4 gene, with *Blautia* genus OUT 194733, and identified as significantly associated with VFM. Several experimental studies suggest that a diet with high fat content may affect epithelial integrity, leading to impaired intestinal permeability and thus systemic inflammation through translocation of Tlr ligands of the Class C G protein-coupled receptors [24,95]. Sprague Dawley rats typically exhibit either obesity-prone or obesity-resistant traits, making them a suitable model to explore the dissociation of nutrition from obesity-related inflammation [96]. In high-fat diet conditions, only obesity-resistant mice display elevated Tlr4 expression linked to ileal inflammation. Furthermore, obesity-prone mice exhibit heightened intestinal permeability and serum endotoxin levels, contrasting with obesity-resistant counterparts. However, both groups experience similar alterations in their microbiota due to the diet. Antibiotic treatment could impact epithelial integrity, as metronidazole reduces mucus thickness, predisposing mice to worsened colitis triggered by *Citrobacter rodentium* [97]. Additionally, the widespread use of antibiotics in early life has been proposed as a factor contributing to the obesity epidemic. Nonetheless, it remains crucial to ascertain whether gastrointestinal barrier function is genuinely compromised in human obesity [98]. In an animal study, it was observed that obese (ob/ob) mice had more Endotoxin-producing Enterobacteriaceae than lean mice with the same diet. The phylum *Bacteroidota* is less abundant, with an increase in phylum *Bacillota*. This suggests that variations in the composition of IM may contribute to obesity, although a causal relationship between these factors and obesity development remains unconfirmed. It has been noted that in obese twin brothers, there is a decrease in the ratio of *Bacillota*/*Bacteroidota* and an increase in the percentage of *Bacillota*, which is associated with the amplification of bacterial genes encoding key enzymes related to carbohydrate metabolism. Thus, these encoded enzymes can consequently increase the ability to digest food and provide energy to the host in the form of SCFAs [99]. Studies have noted that the colonization of intestinal microbiota (IM) in obese mice (and humans) can replicate the obese phenotype compared to germ-free recipient mice. Additionally, obese children exhibit distinct IM compared to lean children, hinting that early childhood microbiota composition may influence obesity later in life, suggesting that diversifying the IM during early life could be an effective obesity prevention strategy. Moreover, a high-fat diet has been shown to alter IM composition in mice [100]. The *Clostridium* genus (such as *Clostridium coccoides*) and *Bifidobacterium* spp. were significantly reduced in obese mice, while the genus ratio *Lactobacillus*/*Enterococcus* and *Bacteroides* were comparable. Indeed, observations indicate that a 14-week high-fat diet induced comparable changes, notably a significant decrease in the *Eubacterium rectale/C. coccoides* ratio of *Bifidobacterium* spp. Interestingly, administering *Bacteroides uniformis* CECT 7771 orally improved immune and metabolic disorders induced by high-fat diet, linked to alterations in IM in obese mice [101]. It has been noted that the mice with obesity-related hepatocellular carcinoma showed that Gram-positive bacteria such as those from the orders *Clostridiales*, *Bacillales*, and *Lactobacillales* were drastically increased. The severity of hepatocellular carcinoma may be heightened by an increased population of Gram-positive bacteria, which produce elevated levels of the IM metabolite deoxycholic acid, potentially damaging host DNA through bile acid metabolism [102]. Oral antimicrobial therapy targeting Gram-positive bacteria significantly mitigated hepatocellular carcinoma severity in high-fat diet-fed mice, suggesting that obesity induced by a high-fat diet correlates with heightened levels of clostridia and bacilli, thereby contributing to hepatocellular carcinoma development via deoxycholic acid production [103]. *Ruminococcaceae* and *Rikenellaceae* families were elevated in high-fat diet-fed mice, and specific alterations in IM composition induced by the high-fat diet contribute to the obese phenotype [102,104]. *Akkermansia muciniphila* populations, notably reduced in congenitally obese and high-fat-diet-induced obese mice, play a role in weight gain and IM composition [105]. This bacterial species, colonizing the mucus layer due to its mucus-degrading ability, was found to be negatively correlated with body weight, Type 1 diabetes, and Type 2 diabetes. Normalizing the *A. muciniphila* ratio in obese mice, either through oral administration or oligofructose treatment, improved various metabolic disorders, including fat gain, metabolic endotoxemia, adipose tissue inflammation, and insulin resistance [106,107]. The beneficial effects observed necessitated viable *A. muciniphila* cells, as heat treatment rendering *A. muciniphila* non-viable did not yield improvements in these metabolic disorders. In obese individuals, the IM appears to trigger chronic low-grade inflammation within the host gut. Chronic experimental metabolic endotoxemia induced by obesity, diabetes, and insulin resistance triggers the expression of various inflammatory factors. Obesity induced in rats via a high-fat diet resulted in alterations in IM composition and activation of Toll-like receptor 4 (TLR4) in the intestinal epithelium [108]. It is hypothesized that activation of the TLR4 pathway through IM changes caused gastrointestinal inflammation associated with the obese phenotype. The association of germ-free mice with *Enterobacter cloacae* that produces endotoxin and *E. cloacae* strain B29, which was isolated from an obese human sample, causes obesity and disorders of glucose homeostasis, when fed a high-fat diet, but not a normal diet [109]. Thus, lowering plasma bacterial-induced endotoxins levels may be a powerful weapon for controlling metabolic diseases. Certainly, the transfer of IM from lean, healthy donors to patients with metabolic syndrome through small intestinal injections enhances insulin sensitivity [110,111]. The enhancement in insulin sensitivity observed in recipient patients correlates with an augmentation in the abundance of bacteria capable of producing butyric acid, indicating a potential role of microbial butyric acid in fostering this improvement [112]. In several studies, an increase in the ratio of *Bacillota/Bacteroidota* is observed in obesity, while weight loss increases *Bacteroides* spp. [32]. But there are also studies that come to contradictory results. The differences in IM (obese, overweight, and normal) were assessed, and, finally, the ratio of *Bacillota/Bacteroidota* shifted in favor of *Bacteroidota* in overweight and obese subjects. It was observed that bacterial diversity was significantly greater in obese subjects than in normal weight subjects, as was the ratio of Bacillota to *Bacteroidota* [113,114]. Gender, BMI, and dietary fiber intake contribute to shaping the IM in humans. *Bacteroidota* populations and biodiversity are lower in overweight and obese women than in normal weight women, controlling for gender. Gender likely affects the IM through differences in intestinal transit time, which appears to be greater in women than men. The obese subjects had lower amounts of *Clostridium perfringens* and *Bacteroides* than normal weight subjects [115]. The obese individuals with low bacterial biodiversity are characterized by a greater degree of obesity, insulin resistance, and dyslipidemia. In addition, a more pronounced inflammatory profile is observed compared to individuals with high IM bacteria diversity and an easier tendency to gain weight [116]. The IM inhibits AMP-activated protein kinase (AMPK) action associated with fatty acid oxidation in liver and skeletal muscle (Figure 8) [36]. In mice inoculated with IM, knock-out mice for the receptors GPR41 and GPR43 (G-protein coupled receptor) to which short-chain fatty acids bind, it was observed that they did not gain weight compared to normal ones as well as vaccinated animals [117]. It is worth capturing and mentioning the role of yeast and fungi in weight changes.

The second hypothesis links obesity to the IM through low-grade inflammation. It is known that obesity and insulin resistance are conditions where low-grade inflammation prevails, characterized by the secretion of pro-inflammatory cytokines (IL-1, IL-6, TNF-α). It appears that the low-grade inflammation induced by a high-fat diet induces, both in experimental animals and in humans, an increase in the concentration of LPS in the bloodstream and can lead to metabolic endotoxemia derived from bacteria (Figure 9) [118]. In animal studies, the subcutaneous administration of lipopolysaccharide (LPS) can lead to an increase in body weight and insulin resistance without affecting energy intake. Endotoxins activate macrophages in adipose tissue and peripheral circulation, which are known to modulate immunity and chronic inflammation [119]. LPS is a constituent of the cell membranes of Gram-negative bacteria like *Bacteroidota*. Alterations in the composition of the IM may potentially lead to an elevation in their concentration. They are transported from the large intestine via chylomicrons into the systemic circulation. Chylomicrons (lipoproteins) are synthesized by intestinal epithelial cells in response to a high-fat diet. The binding of LPS to CD14 cells via the TLR4 receptor causes the secretion of pro-inflammatory compounds. Mice genetically engineered to lack Toll-like receptor 4 (TLR4), which detects LPS, demonstrate resistance to diet-induced obesity and insulin resistance [120].

In a study recording daily food intake, it was observed that in male adolescents, cytokines were associated with total lipid consumption. The inclusion of LPS into lipoproteins elevates the lipid fraction that triggers inflammation. This could be mediated by the CD36 and the scavenger receptor class B type I (SR-BI), which have a high affinity for LPS [121]. LPS-incorporated lipoproteins have increased transcytosis through endothelium, and thus translocate a higher number of LPS-incorporated lipoproteins into adipocytes, which may increase adipogenesis and exposure of adipocytes and macrophages to LPS. This contributes to the transition from M2 to M1 macrophages in adipose tissue and adipocyte demise. LPS plays a role in obesity development by directly influencing lipid transportation and storage in adipose tissue [121].

### 4.2. Non-Alcoholic Fatty Liver Disease (NAFLD)

NAFLD is identified by the accumulation of fat (steatosis) in the liver, not attributed to excessive alcohol intake, and impacts approximately 25% of the global population. It is observed in conditions of insulin resistance, but also in metabolic syndrome. NAFLD can progress to liver cirrhosis and hepatocellular carcinoma. Nonalcoholic steatohepatitis (NASH) is the most severe form of NAFLD, and 20% of simple steatosis patients will progress to nonalcoholic steatohepatitis (NASH), making it a leading cause for cirrhosis [122]. Multiple genetic, metabolic, inflammatory, and environmental factors are suggested to play a role in the development of the condition. In vivo research indicates that conventional mice exhibit higher liver triglyceride levels compared to germ-free mice, despite consuming less food. IM colonization correlates with increased absorption of monosaccharides from the intestinal lumen, stimulating de novo fatty acid synthesis and liver triglyceride production. This is supported by heightened enzyme activity like acetyl-CoA carboxylase and fatty acid synthesis [123].

The presence of microorganisms’ fermentation byproducts, such as ethanol in the gut, is a significant contributor to obesity development in mice, and could be linked to the onset of NAFLD. Like obesity, the chronic inflammation triggered by microbial endotoxins involves CD14-TLR4 signaling and activates hepatic Kupffer cells in mice, thus contributing to NAFLD pathogenesis [124,125]. Moreover, IM can indirectly induce liver steatosis and lipoperoxidation through farnoid X receptor-mediated signaling, affecting bile acid secretion in mice [126]. Furthermore, the IM can induce steatosis through increased absorption of monosaccharides, production of hepatotoxic ethanol, chronic low-grade inflammation induced by microorganism’s endotoxins, and modulation of bile acid metabolism. Mice with nonalcoholic fatty liver disease induced by a high-fat diet showed a change in the gut microbiome due to intestinal barrier destruction and increased intestinal permeability, accompanied by detectable intestinal bacteria in the liver [126]. IM overgrowth in obese patients may be associated with steatosis [127]. Patients with NASH have IM dysbiosis with systemic inflammation and elevated serum levels of tumor necrosis factor alpha (TNF-a) and breakdown of intestinal mucosal tight junctions [128,129]. In addition, systemic higher ethanol levels in NASH patients indicate that the growth of ethanol-producing bacteria may be involved in pathogenesis, and chronic endotoxemia correlates with the severity of NAFLD [130].

Reduced intake of choline in one’s diet could potentially contribute to the emergence of NAFLD in humans. In an experiment, 15 female participants adhered to carefully regulated diets where choline levels were modified accordingly [131]. Dietary choline deficiency led to dysbiosis, and levels of the *Gammaproteobacteria* class and *Erysipelotrichia* phylum were positively correlated with changes in liver fat content. Individuals diagnosed with NASH exhibited a reduced presence of the bacterial genus *Bacteroidota* in comparison to both healthy individuals and those with uncomplicated steatosis. This resemblance to the intestinal microbiota profile seen in obese individuals hints at potential diagnostic and therapeutic avenues. These findings propose that disparities in intestinal microbiota among NAFLD, NASH, obese, and healthy individuals could serve as diagnostic indicators and targets for preventative or therapeutic interventions, such as probiotic treatments [132].

### 4.3. Atherosclerosis

Cardiovascular disease encompasses multiple nosologically entities, such as coronary heart disease, stroke, arterial hypertension, and heart failure. Clinical manifestations of cardiovascular disease are often preceded by subclinical disorders in the vessels, including endothelial dysfunction and arterial stiffness [133]. Atherosclerotic vascular disease can be influenced by genetic and environmental factors (such as diet) and is associated with the IM. The presence of high bacterial diversity in atherosclerotic plaques and the connection between microbiota and plaque stability have been confirmed. The development of atherosclerosis involves an initial damage to the endothelial cells, subsequent accumulation of lipids, and adherence of macrophages and other immune cells to the artery walls [134,135]. Atherosclerosis encompasses metabolic and inflammatory elements, both potentially affected by changes in the IM. The connection between IM and atherosclerosis was initially recognized through the detection of various bacterial DNA types within atherosclerotic plaques [134]. The *Chryseomonas* genus was present in all examined human atherosclerotic plaques, and Streptococcus was present in most samples. The genus *Collinsella* was increased in patients with symptomatic atherosclerosis. Many bacterial types detected in atherosclerotic plaques were also present in samples from the oral cavity and IM of the same individuals. This implies that these distant microbial communities could serve as the primary origin of bacteria associated with atherosclerosis [136]. Choline, betaine, and trimethylamine N-oxide (TMAO) are phospholipid-related molecules indicated by analysis of plasma metabolites, and appear to promote atherosclerosis, serving as biomarkers to predict cardiovascular disease risk [137]. Indeed, a study using apoE-deficient mice as a model of atherosclerosis showed that plasma TMAO levels are positively correlated with the area of aortic damage. Eating patterns can influence both the makeup of the IM and its capacity to metabolize trimethylamine (TMA) and TMAO derived from dietary L-carnitine [138,139]. Dietary L-carnitine is metabolized to TMA by IM and further converted to TMAO in the liver, accelerating atherosclerosis in mice. The expression level of hepatic flavin monooxygenases, which converts trimethylamine (TMA) to TMAO, is linked to plasma TMAO levels. Finally, the genes encoding peptidoglycan synthesis were increased, while phytohydrogenase was completely decreased in patients with atherosclerosis. Symptomatic patients with atherosclerosis exhibited decreased serum levels of β-carotene, indicating that the inflammatory condition associated with atherosclerosis may correlate with distinct alterations in the IM (Figure 10) [140].

There are numerous species in the IM that produce trimethylamine, including *Anaerococcus hydrogenalis*, various species such as *Clostridium sporogenes*, *Enterocloster asparagiformis*, *Edwardsiella tarda*, *Hungatella hathewayi*, *Escherichia fergusonii*, *Proteus penneri*, and *Providencia rettgeri*. These bacteria can metabolize choline into trimethylamine (TMA), which is subsequently converted into trimethylamine-N-oxide (TMAO) by hepatic monooxygenases in the liver [141]. TMAO is implicated in the adverse effects of dysbiosis on atherosclerosis and cardiovascular disease, serving as a reliable predictor of negative cardiovascular events. Its detrimental mechanisms involve triggering inflammatory signals, inhibiting reverse cholesterol transport, and increasing Ca^2+^ ion release in platelet cells [113,114]. In chronic kidney disease, TMAO can enhance the expression of bone-related molecules, activate NLRP3 and NF-κB, and promote calcification of smooth muscle cells and vascular tissue. Elevated TMAO levels can also induce the expression of the transcription factor Foxo1, leading to hyperglycemia [142,143].

### 4.4. Type 1 and 2 Diabetes Mellitus

Certain bacterial families have been observed to potentially confer protection against Type 1 diabetes. Research suggests that the administration of antibiotics in experimental animals can prevent the onset of Type 1 diabetes mellitus [143]. Exposing experimental animals to *Mycobacterium* and *Streptococcus* spp. during incubation shielded them from diabetes development. Apart from increased intestinal permeability, individuals with Type 1 diabetes mellitus exhibit elevated inflammatory cells in the gut and decreased CD4, CD25, and T-cells in the immune system [144]. Another study revealed that individuals who developed Type 1 diabetes mellitus had higher levels of *Bacteroidota* and lower levels of *Bacillota* compared to healthy controls. Furthermore, those who developed Type 1 diabetes mellitus were colonized with fewer bacteria compared to healthy controls [141].

The Type 2 diabetes mellitus is characterized by reduced insulin secretion from the β-cells of the pancreas, increased insulin resistance, and disruption of incretin secretion. IM dysbiosis in Type 2 diabetes mellitus is likely promoted by diet-induced obesity and corresponding metabolic complications through a variety of mechanisms, including immune regulation, differential energy regulation, altered regulation of gut hormones, and proinflammatory mechanisms (such as lipopolysaccharide endotoxins) [142]. There is a reduction in *Bacteroides*/*Prevotella* spp. ratio which is associated with obesity and metabolic disorders. A recent study about IM dysbiosis found that *Blautia*, *Odoribacter*, *Oscillibacter*, and *Pseudoflavonifractor* were significantly positively associated with insulin resistance. Among these, *Blautia* is also considered to be positively associated with impaired glucose tolerance in Type 2 diabetes mellitus, and its abundance appears to be reduced after gastric bypass surgery. Individuals with Type 2 diabetes had an increase in the number of various opportunistic intestinal pathogens as *Escherichia coli*, some of the species of *Clostridium*, *Bacteroides caccae*, and *Eggerthella lenta*, while there was a decrease in the number of hydroxybutyric acid-producing bacteria (such as *Roseburia* spp., *Faecalibacterium prausnitzii*, *Eubacterium eligens*, and *Bacteroides intestinalis*) [143]. The latter indicates the potential impact of butyric acid-producing bacteria and, therefore, the relationship of the microbiota to insulin resistance and metabolic homeostasis of Type 2 diabetes mellitus. Hydroxybutyric acid is the main source of energy for maintaining the function of the cells of the digestive system. In the large intestine, hydroxybutyric acid is mainly produced by the bacteria *C. coccoides* and *Eubacterium rectale*. Indeed, a series of fecal transplant experiments demonstrated that the IM plays an important role in energy absorption, adipose tissue accumulation, and insulin resistance [144].

However, changes in the IM were found in colon cancer patients and in elderly subjects, suggesting that hydroxybutyric-acid-producing bacteria could potentially have a protective role in the functioning of the IM [101]. The IM interact through the axis of the immune-metabolic system with various mechanisms and can lead to certain effects on health. In type 1 diabetes and intestinal dysbiosis, increased gut permeability and altered immune-regulated mechanisms appear to trigger the autoimmune response that leads to cell destruction in the pancreatic islets. In type 2 diabetes, saturated fat and dysbiosis due to the “obesogenic” diet cause inflammation and alterations in intestinal permeability that contribute to the onset of the disease. In Type 2 diabetes mellitus, the increased bacteria are mainly pathogenic, such as the *Streptococcus mutans.* Furthermore, species from the *Lactobacillaceae* phylum (such as *Lactobacillus gasseri*) are also predominantly abundant [145]. This indicates, in IM dysbiosis, that no single or small group of bacteria may be fully responsible for insulin resistance or Type 2 diabetes mellitus and its complications, making Type 2 diabetes mellitus once again a multifactorial disease. Obesity is clearly a major risk factor for Type 2 diabetes [146].

Thus, the main biomolecular mechanisms of action of the IM and metabolites derived from obesity and related metabolic dysfunctions (insulin resistance and Type 2 diabetes mellitus) involve the IM’s contribution to the hydrolysis of complex polysaccharides from dietary fiber. Therefore, it could contribute to increasing energy conservation and produce SCFAs such as acetate, propionate, and butyric acid. These SCFAs affect host metabolism by different pathways. SCFAs can activate the G protein-coupled receptor (Gpr) 41, which induces the expression of the intestinal hormone peptide YY, affecting intestinal motility and potentially leading to an increase in peristalsis and reduced dietary energy conservation. SCFAs can also activate Gpr43 and Gpr41, inducing glucagon-like peptide-1 (GLP-1) secretion, increasing insulin sensitivity, and inducing satiety [147]. Butyric acid provides energy to enterocytes, influencing food intake and causing the synthesis of GLP-2, thereby strengthening the function of the intestinal barrier. Therefore, pathogens, proinflammatory cytokines, and harmful metabolites more easily pass through the intestinal vascular barrier and enter the circulatory system [109,148].

Hyperglycemia in patients with Type 2 diabetes mellitus has been found to inhibit the integrity of the intestinal barrier and adhesive junctions, accelerating their destruction. Several studies showed that the dysbiosis of intestinal bacteria plays a key role in the destruction of the intestinal barrier. A study in normally fed leptin-deficient mice demonstrated that increased intestinal permeability to lipopolysaccharide resulted in a change in the proportion of Gram-negative bacteria in the intestinal lumen, correlating with the presence of insulin resistance [149].

Butyric acid may facilitate the development of peripheral regulatory T cells (Treg) by inhibiting the deacetylation of histones 6 and 9, leading to histone H3 acetylation. This process enhances the expression of the Treg-specific forkhead transcription factor FoxP3 [150]. Intestinal dysbiosis, potentially influenced by “obesogenic” diets high in saturated fat, can foster the proliferation of potential pathogens such as Gram-negative bacteria and LPS derivatives. These pathogens exert pro-inflammatory effects by generating cytotoxic compounds like H2S or interacting with innate immune receptors (TLR4, TLR2). Consequently, this contributes to the production of inflammatory cytokines, recruitment of inflammatory cells, and translocation of bacterial products (LPS, DNA) via intercellular and paracellular pathways, activating inflammation in peripheral tissues [151,152]. Finally, advanced glycation end products (AGEs) which are produced in parallel with the progression of Type 2 diabetes mellitus, participate in the development of diabetic complications. In addition to endogenous formation, AGEs can also accumulate in foods, especially heat-treated diets. Intake of foods rich in AGEs can reduce the diversity and change the composition of the IM [153,154,155]. It has been observed that a mice diet high in AGEs significantly increased the abundance of *Helicobacter* spp., which stimulates vascular inflammation and atherogenesis and decreased the “good bacteria” which help the host resist infection by pathogens, inflammation, and oxidative stress such as *Lachnospiraceae* NK4A136, *Roseburia butyrates*, and those from the genus *Alistipes* [156,157].

## 5. The Perspectives from the Individual Research Metabolome to Personalized Therapy

### 5.1. Nutritional Interventions

Dietary modifications are deemed essential for the prevention and treatment of features associated with metabolic diseases. A significant limitation of the global dietary recommendation is a necessity to consider intrinsic properties of foods, such as the glycemic index, which estimates the postprandial glycemic response to specific foods [158]. In a study, marked differences in postprandial glycemic response were observed in both standardized and real-life meals, such as white bread, with many participants exhibiting different responses to the same foods [158,159]. These findings raise concerns about the global dietary recommendations based solely on food properties. Therefore, understanding the factors driving interindividual differences in response to food is crucial for enhancing personalized nutrition and prevention of MetS. Recent studies suggest that this approach can yield long-term results if tailored individually to each person. Although the integration of personalized intestinal IM parameters in the diagnosis and nutritional planning for individuals predisposed to or suffering from metabolic diseases is considered attractive for clinical research, it is still in its early stage. Questions persist regarding the long-term effectiveness of this approach, warranting future human-based studies [159,160,161,162,163].

### 5.2. Faecal Microbiota Transplantation (FMT)

FMT involves transferring purified microbes from the feces of a healthy donor into a person with conditions related to IM, such as obesity or diabetes, with the aim of correcting or replacing the pathological microbiota [111,164]. Indeed, studies have shown that after transplanting fecal microbiota from healthy mice into mice fed a high-fat diet, the intestinal barrier was restored, and metabolic disorders improved. A promising study in metabolomics diseases demonstrated that microbiome transplantation from lean donors in obese subjects improved insulin sensitivity in patients, accompanied by changes in the IM, including the expansion of butyric acid producers [165]. However, the effectiveness of this approach in addressing conditions responsible for metabolic diseases requires additional documentation from long-term clinical studies. One potential drawback of this method is the uncertainty surrounding the ability of the transplanted microbiome to modify the existing pathological microbial composition effectively. Therefore, it is probable that factors contributing to intestinal microbiota (IM) dysbiosis in an individual, such as genetics and lifestyle, may persist even after Fecal Microbiota Transplantation (FMT). This persistence can resist or reverse microbial changes, potentially causing a return to the diseased state [166].

### 5.3. Probiotic and Prebiotic Formulation in Metabolic Diseases

Probiotics and prebiotics have been introduced into our lives as health-enhancing supplements, and numerous publications have highlighted their effects, such as improving the intestinal environment, regulating immune system functions, and preventing pathogenic microbial infections. The action of probiotics and prebiotics against obesity is well recognized. For instance, *Lactobacillus acidophilus* La5 and *Bifidobacterium animalis* Bb12 can produce conjugated linoleic acid (CLA), a naturally occurring isomer found in ruminant dairy products, which has been shown to help prevent conditions like colon cancer, atherosclerosis, and obesity in mice [167,168]. Probiotics contain non-pathogenic live microorganisms that modify the host’s IM. Studies have demonstrated the effects of oral administration of probiotics in obese 4 non-diabetic mice. At 32 weeks of age, only 21% of the probiotic-treated mice developed diabetes compared to 81% in the phosphate-treated control group. The first group also showed reduced inflammation of the pancreatic islets and a decreased rate of β-cell destruction [169]. Additionally, a milk fermentation product containing probiotic bacteria significantly delayed the onset of glucose intolerance, hyperglycemia, hyperinsulinemia, and dyslipidemia, and reduced oxidative stress in diabetic rats fed a high-fructose diet [170]. A TEDDY (The Environmental Determinants of Diabetes in the Young) study revealed that the administration of probiotics to children under three months of age who had an increased genetic risk of T1DM led to a 33% reduction in pancreatic β-cell autoimmunity [171]. Probiotic strains that produce conjugated linoleic acid (CLA), such as *Lacticaseibacillus rhamnosus* PL60 and *Lactiplantibacillus plantarum* PL62, have been shown to diminish body weight gain and white adipose tissue mass in mice fed a high-fat diet. These effects were observed without altering food intake. Additionally, *Lactiplantibacillus plantarum* PL62 reduced body weight and glucose levels in high-fat-diet-induced obese mice. Probiotics are reported to reduce adipocyte size in various fat deposits, a crucial indicator of anti-obesity probiotics. Possible mechanisms involve increased excretion of neutral sterols and bile acids in feces, coupled with a reduction in the absorption of triglycerides, phospholipids, and cholesterol through the lymphatic system [172]. In experimental study with pre-adipocyte cell lines, supplementation with probiotics and prebiotics in a reduction of adipocyte differentiation, contributing to decreased in a reduction of adipogenesis and adipocyte differentiation, contributing to decreased fat accumulation [173]. Furthermore, administering a particular probiotic strain to mice resulted in elevated serum levels of angiopoietin type 4, which acts as a lipoprotein lipase inhibitor, thereby regulating fat accumulation in adipocytes. Additionally, supplementation led to the upregulation of insulin-sensitizing hormones such as adipsin and adiponectin [174]. Furthermore, supplementation with a specific probiotic strain, such as *Limosilactobacillus reuteri* ATCC4659, in mice resulted in decreased body weight and liver fat, along with increased expression of palmitoyl carnitine transferase 1A in the liver, suggesting the activation of hepatic β-oxidation [174].

Modulation of the IM probiotic treatment in obese mice has been observed to act favorably on the intestinal barrier, reducing metabolic inflammation induced by lipopolysaccharides. Elevated levels of Bifidobacterium spp. have been found to alter the inflammatory response in obese mice by boosting the production of Glucagon-Like Peptides (GLPs) while decreasing intestinal permeability [175]. The rise in Bifidobacterium spp. due to probiotic therapy likely correlates with increased secretion of gut peptides GLP-1 and YY, which help reduce insulin resistance and enhance β-cell function. Additionally, probiotic treatment increases GLP-2 levels in the colon, enhancing intestinal barrier function and ultimately lowering plasma lipopolysaccharide levels. However, there is limited research addressing changes in microbial composition induced by probiotic supplementation regarding anti-obesity effects [176].

Supplementing mice with *Lacticaseibacillus rhamnosus* GG and *Latilactobacillus sakei* NR28 led to a reduction in the prevalence of *Bacillota* and *Clostridium* cluster XIVa in the small intestine. This reduction resulted in decreased weight gain, fat mass, and expression of lipogenic enzymes such as hepatic steatoyl-CoA desaturase-1, fatty acid synthase, and acetyl-CoA carboxylase [177]. However, when mice were supplemented with *L. acidophilus* NCDC13 in a diet-induced obesity model, there was an increase in the total number of Bifidobacteria in cecal and fecal contents without a reduction in fat deposition [178]. In another study, oral inoculation with *Lactobacillus ingluviei* increased the total amount of fecal *Bacillota* and *Lactobacillaceae* phyla in mice, resulting in increased body weight, liver weight, and metabolism. In overweight subjects, oral administration of *Lactobacillus gasseri* SBT2055 reduced abdominal visceral fat and subcutaneous fat [179]. Supplementation of *L. rhamnosus* GG in infant formula for six months resulted in better growth, but also higher weight gain. However, prenatal, and postnatal administration of *L. rhamnosus* GG prevented excessive weight gain in children. The physiological effects of probiotics in human samples appear to be varied. Oral administration of *Clostridium butyricum* MIYARI 588, a butyric-acid-producing anaerobe, reduced the progression of non-alcoholic fatty liver disease (NAFLD) in mice with diet-induced steatosis [180]. Supplementation with the “De Simone Formulation” (DSF), a highly concentrated probiotic supplement with eight different strains of “good” bacteria, improved insulin sensitivity and reduced hepatic adipogenesis in leptin-deficient ob/ob mice [181]. In apolipoprotein E-deficient mice, DSF improved insulin resistance, prevented histological changes in mesenteric adipose tissue, tissue inflammation, steatohepatitis, and reduced the extent of aortic plaques [182]. Another study using DSF in the liver of young mice with inflammatory and oxidative damage induced by a high-fat diet found that the probiotic prevented the increase in inflammatory markers compared to the control group. Overall, these studies suggest that various probiotics may alleviate fatty liver disease, at least at the preclinical level [183].

Prebiotic carbohydrates reduced systemic endotoxin levels and the expression of inflammatory cytokines in the liver [184]. The improvement of metabolic inflammation in obese mice may be attributed not only to alterations in the microbiota, but also to the expression of glucagon-like peptide 2 (Glp2), a gut growth factor with anti-inflammatory properties that enhances intestinal barrier function [184,185]. Prebiotic therapy enhanced intestinal permeability, reduced systemic inflammation, lowered hepatic expression of pro-inflammatory cytokines, and improved insulin sensitivity in obese ob/ob mice, while also increasing intestinal Glp2 expression. Similarly, treatment with a Glp2 agonist demonstrated comparable beneficial effects. Finally, the endocannabinoid system has been shown to mediate the effects of the IM on mucosal permeability and may also play a key role in the regulation of insulin and obesity (Figure 11) [186,187].

Among the extensively researched probiotic supplements, inulin extracted from plants, and related compounds like fructooligosaccharides, varying in fructose polymerization levels, have garnered significant attention. Inulin notably fosters the proliferation of bifidobacteria, linked to diminished weight gain, enhanced glucose regulation, and mitigation of obesity-related inflammation, commonly referred to as metabolic endotoxemia [188]. Sequencing the intestinal bacteria of ob/ob mice fed prebiotic oligofructose revealed changes in the IM involving more than 100 species, with 16 showing more than a 10-fold changes in abundance. Among those identified was *A. muciniphila*, which negatively correlates with body weight [188,189]. An important effect of inulin supplementation appears to be the influence on the production of gastrointestinal hormones (such as GLP-1, PYY, ghrelin, and other) through microbial alterations in both animals (rats) and humans [190]. These hormones govern diverse physiological processes like insulin release via incretin and gastrointestinal movement, implying their involvement in the anti-obesity effects of prebiotics. The microbial generation of short-chain fatty acids (SCFAs) has been proposed as crucial in stimulating the secretion of gut hormones like GLP-1 [191]. Other studies have demonstrated that prebiotic fiber reduces the ratio of *Bacillota/Bacteroidota* in obese rats and improves NAFLD by reducing hepatic de novo lipogenesis [192,193]. The addition of the fungal glycan chitin increases the number of bacteria associated with the *Clostridium* XIVa complex, including *Roseburia* spp., which is accompanied by reduced weight gain and fat development [194]. Wheat-derived arabinoxylans have been reported to restore the ratio of *Bacteroides/Prevotella* spp., as well as *Roseburia* spp., while significantly increasing the number of *Bifidobacterium* spp., specifically *Bifidobacterium animalis lactis*, in the caecum of mice fed a high-fat diet [176]. Dietary supplementation with inulin increases *Bifidobacterium* spp. and *F. prausnitzii*, while reducing *Bacteroides intestinalis*, *Bacteroides vulgates*, and *Propionibacterium* in obese women. Additionally, consumption of galactooligosaccharides for 12 weeks increased several types of *Bifidobacterium* spp. and reduced the number of *Bacteroides* in healthy individuals [195,196].

### 5.4. Antibiotics

Antibiotics can reduce systemic endotoxin levels and the expression of inflammatory cytokines in the liver. In a study where norfloxacin and ampicillin (1 g/L) were administered for two weeks to ob/ob mice (obese mice with hyperinsulinemia and dyslipidemia), suppression of aerobic and anaerobic cecal bacteria was observed, accompanied by a significant improvement in postprandial glycemia and glucose tolerance [197]. Both plasma levels of lipopolysaccharides and expression of TNF-α in the jejunum were significantly lower in the antibiotic-treated mice group than in the control group, suggesting that modification of the IM by the combination of norfloxacin and ampicillin improved the inflammatory state of the intestine. Similar results when were observed when mice were administered polymyxin B and neomycin [198]. The first therapeutic attempt aimed at modifying the human microbiota was made in a pilot study in 28 people with pre-diabetes or diabetes, who were given the agent NM504 versus a placebo for 28 days. NM504 is a combination of insulin with β-glucan and anthocyanin rich in polyphenols. This combination alters the IM, strengthening a series of beneficial microbes against the development of metabolic diseases, at the expense of another series of microbes that contribute pathogenically to the development of these diseases. After one month of treatment, a decrease in glucose metabolism, HbA1C, C-reactive protein, total cholesterol, and gut pH was observed, while IgA levels increased [197,198].

## 6. Conclusions

Recent studies have highlighted the potential role of IM in the pathogenesis of these metabolic disorders. Thus, MetS can be associated with dysbiosis of the IM which influence the body’s homeostasis through dysregulation of immunity and dysmetabolism. The MI shows significant variations between individuals being a “fingerprint” for everyone, variable during the years of life (age, lifestyle, medications, the time of day in which a sample is collected, etc.), making its accurate characterization complicated. Therefore, any intervention to prevent and slow the manifestation of such metabolic diseases requires long-term tests based on the IM.

We note that the composition of the IM influences the food derivate energy management by the host and vice versa, thus participating in metabolic regulation. In fact, some genera of microorganisms (such the *Bacillota* genus) have a greater ability to obtain energy from foods. Bacterial fermentation of foods produces substances (e.g., butyric acid) which are sources of energy. Microbial derivatives such as SCFAs influence the expression of important peptides that regulate metabolism (glucagon-like peptide 1 and peptide YY).

Animal studies have demonstrated that the IM influences the host’s energy metabolism, suggesting that certain bacterial targets involved in controlling MetS-related diseases can be identified. Many studies have reported differences in IM composition between obese and lean individuals, both in animal models and in humans. However, it is premature to conclude that specific genera, classes, or species of the IM microorganisms are consistently associated positively or negatively with the obese phenotype. The research findings suggest that higher microorganism biodiversity plays a beneficial role in preserving healthy body weight and glucose balance. Conversely, decreased diversity is linked to conditions like inflammation and the rising incidence of obesity and its associated conditions, including diabetes, atherosclerosis, and NAFLD. Elevated serum non-healthy metabolites can be lead to a metabolic endotoxemia state (intestinal and systemic), such as the LPS observed in obese individuals, indicating that specific microbial components of the intestine may contribute to metabolic disorders. But what about the if we have a higher amount of those which are healthy? Hence, further controlled investigations involving both humans and animals are imperative to elucidate this intricate relationship. Integrating metagenomic, transcriptomic, and metabolic analyses could offer deeper insights into the molecular mechanisms underlying metabolic interactions between the IM and host physiology.

Metabolic endotoxemia negatively affects IM axes, such as intestine/lung axis, which could have implications for prolonged prognosis and rehabilitation procedures such as those in pneumology. However, the molecular mechanisms between IM-axes (such as oral, lung, liver, skin, and others) and MetS underlying the influence of the intestinal microbiota on host metabolism need to much more in-depth analysis.

More investigation is needed to precisely delineate the state of epithelial integrity in human obesity and its possible implications for the IM.

The potential contribution of the IM to several disease pathogenesis and clinical manifestations such as those of MetS, coupled with its variability, renders it an appealing therapeutic target for diagnosing and treating features of metabolic disease. Dietary habits and lifestyle are recognized as key factors influencing the development and progression of MetS. Therefore, IM-based personalized diets should not only consider how the IM metabolome mediates the effect of diet on host metabolism, but also how diet may influence microorganism biodiversity and, consequently, other health characteristics of the host. This approach, combined with studies utilizing appropriate animal models, will contribute to a better understanding of the function of distinct microbial groups or individual species of the IM and to evaluate the efficacy of therapies, such as probiotics and prebiotics, in controlling MetS-related diseases.

## Figures and Tables

**Figure 1 ijms-25-02841-f001:**
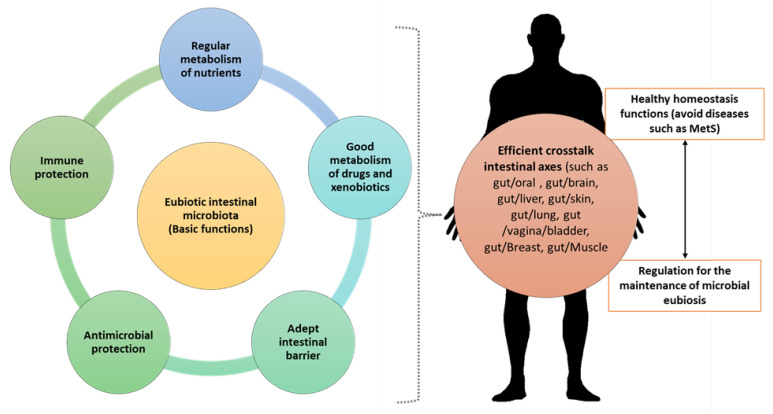
The consequence for the host of achieving a “good balance” (eubiosis) of quantitative and qualitative microorganisms in the IM is significant. This condition can establish a “healthy main functions cycle” that promotes balance in the crosstalk axes, thereby helping to prevent MetS. (Original figure by I.A. Charitos).

**Figure 2 ijms-25-02841-f002:**
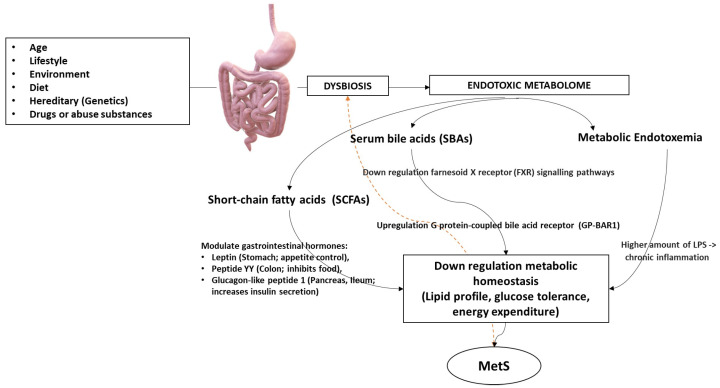
The main key factors involving the IM metabolome and MetS include dysbiosis in bacteria populations. In subjects affected by MetS, the *Lactobacillaceae* family, *Sutterella*, and *Methanobrevibacter* spp. Were observed, while *Akkermansia*, *Odoribacter*, and *Bifidobacterium* spp. were associated with healthy individuals. Endotoxic metabolome throughout the metabolism of SBAs, the production of SCFAs with the metabolic endotoxemia, can lead to the disruption of lipids, altered glucose homeostasis, affect satiety, and create an inflammatory chronic condition. All this can lead to MetS, which in turn further alters the IM dysbiosis. (Original figure by I.A. Charitos).

**Figure 3 ijms-25-02841-f003:**
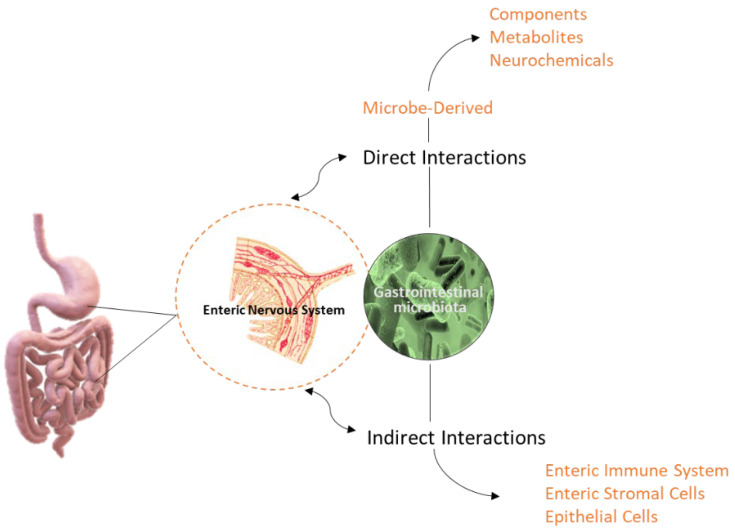
The importance of the Enteric Nervous System (ENS) in the interaction with the IM is significant. The connections between the IM and ENS can manifest through both direct and indirect mechanisms. Bacterial components from Gram-negative bacteria (such as LPS or polysaccharide A) or Gram-positive bacteria (such as peptidoglycan), known as microbe-associated molecular patterns (MAMPs) are detected by receptors expressed in myenteric neurons, enteric glial cells, and innate immune cells. This recognition occurs via surface transmembrane pattern (PRR) or Toll-like endosomes (TLR). Furthermore, neuronal signaling and metabolic products of the IM, such as SCFAs, which are involved in maintaining ENS homeostasis, stimulate various G protein-coupled receptors (GPCRs), PNS, and block the action of histone deacetylases (HDACs). Some bacteria within the IM can release neurotransmitters produced (such as dopamine, serotonin, etc.) that modulate intestinal secretion and motility, thus establishing an axis of interaction between bacteria, neurons, the ENS, PNS, and CNS. (Original figure by I.A. Charitos).

**Figure 4 ijms-25-02841-f004:**
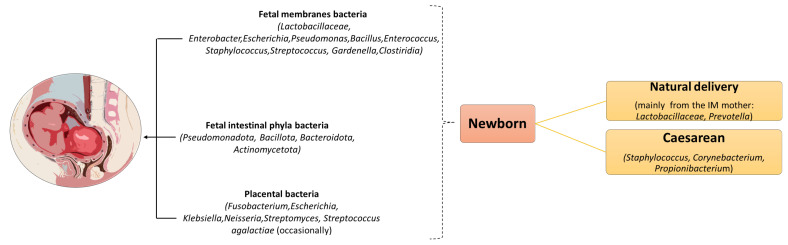
Some key bacteria phyla and species found during pregnancy influence fetal intestinal microbiota. In the early stages of life, the qualitative and quantitative composition of microbial populations depends on the mode of delivery. Individuals delivered via cesarean section have exhibited a heightened susceptibility to allergic conditions and a greater inclination toward developing various diseases overall. (Original figure by I.A. Charitos).

**Figure 5 ijms-25-02841-f005:**
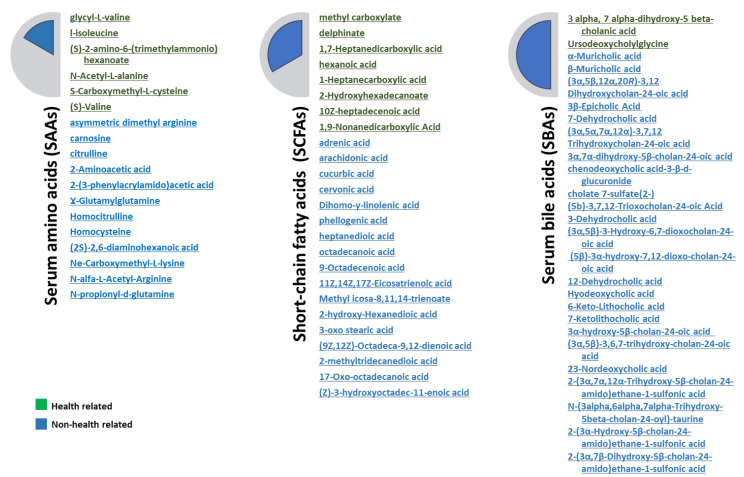
Metabolites produced by the IM, including those associated with favorable body health (highlighted in green) and those correlated with organic and mental ailments (highlighted in azure), play a pivotal role in regulating various aspects of host metabolism and physiology. Microbial products, like short-chain fatty acids (SCFAs), interact with G-protein-coupled receptors (GPCRs) on intestinal epithelial cells (e.g., Gpr41 and Gpr43), influencing energy balance and modulating the release of the gut hormone PYY, as well as regulating the host’s inflammatory response. Activation of TLR5 (e.g., via bacterial flagellum) potentially impacts the composition of the intestinal microbiota, thereby influencing appetite, weight gain, and insulin sensitivity through mechanisms that are not yet fully understood. Bacterial signals also regulate the release of fasting-induced adipose factor (FIAF) from intestinal epithelial cells, which inhibits LPL and thus controls peripheral fat storage. Moreover, the intestinal microbiota modulates energy homeostasis in the liver and muscles, possibly through the phosphorylation of AMP-activated protein kinase (AMPK), although the exact mechanisms remain unknown. GLP-2 supports epithelial barrier function, and a compromised barrier may expose and activate myeloid cells in response to microbial signals such as the endotoxin ligand TLR4. (Original figure by I.A. Charitos).

**Figure 6 ijms-25-02841-f006:**
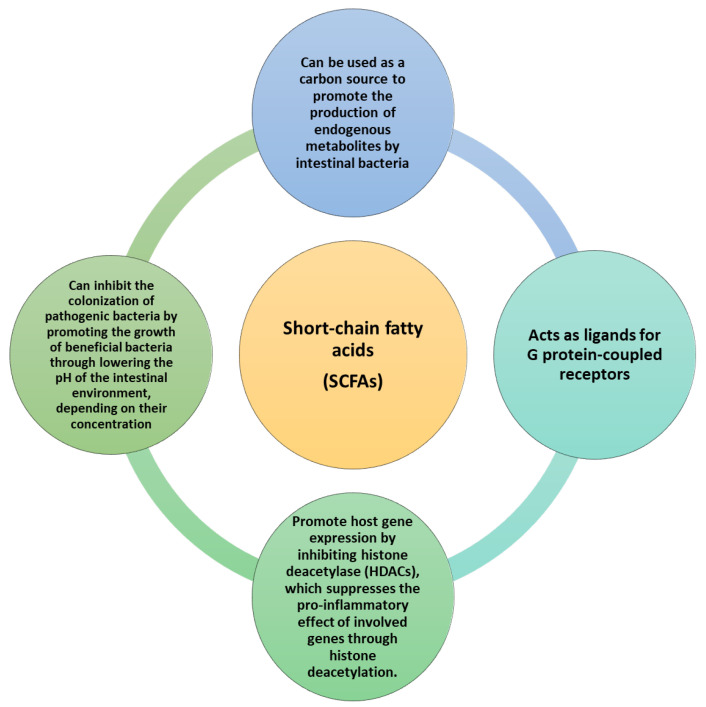
Short chain fatty acids-SCFAs beneficially affect the health of the host in the various modality. (Original figure by I.A. Charitos).

**Figure 7 ijms-25-02841-f007:**
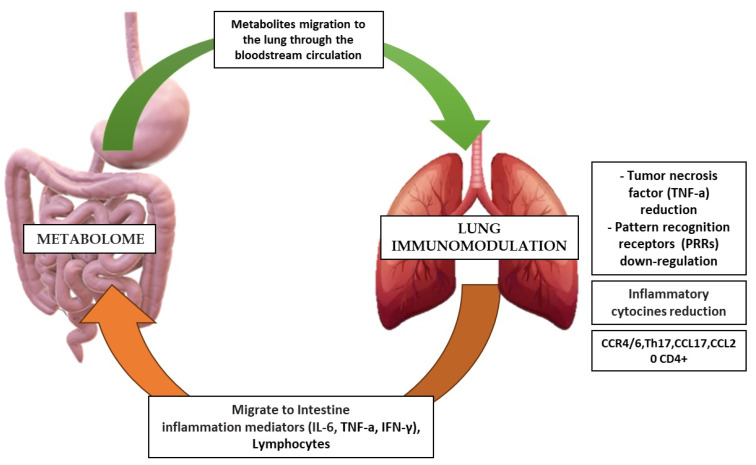
The hypothesis suggests that the IM can modulate the immunological activity of the lung: Lipopolysaccharides (LPS) bind to Toll-Like Receptor (TLR) on the intestinal mucosa, activating dendritic cells that promote the activation of various T cells, particularly T-reg, T-h17, and Th1, which later migrate to the lung through the circulatory stream. Bacterial metabolites (such as SCFAs) directly act on the nuclear factor kappa-light-chain-enhancer of activated B cells (NF-kB), reducing the production of Tumor Necrosis factor (TNF-α) and downregulating pattern recognition receptors (PRRs), resulting in reduced production of inflammatory cytokines: IL-1, IL-12, IL-18, TNF-α, interferon gamma (IFNγ), and granulocyte-macrophage colony-stimulating factor (GM-CSF). (Original figure by I.A. Charitos).

**Figure 8 ijms-25-02841-f008:**
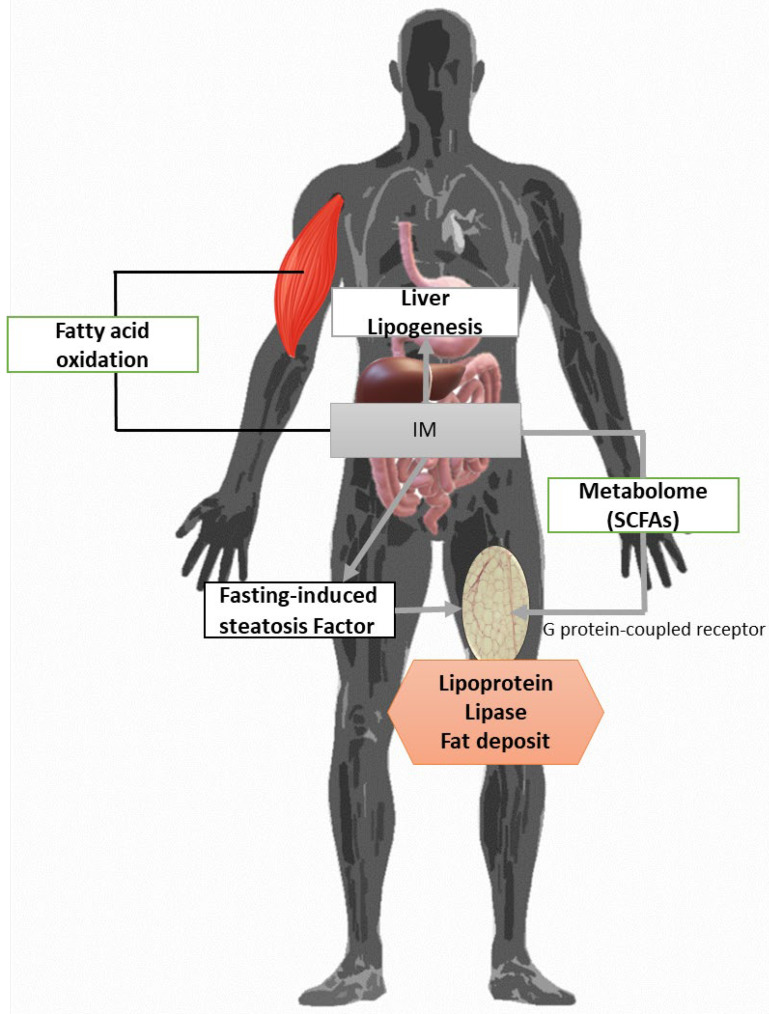
Proposed mechanism for the role of the IM in energy production and fatty storage in the host. After fatty acid oxidation, the biomolecular connection with the IM causes the deposition of fat through three mechanisms: (a) stimulation of lipogenesis in the liver, (b) via the bacteria metabolite SCFAs, and (c) via fasting-induced steatosis factors. (Original figure by I.A. Charitos).

**Figure 9 ijms-25-02841-f009:**
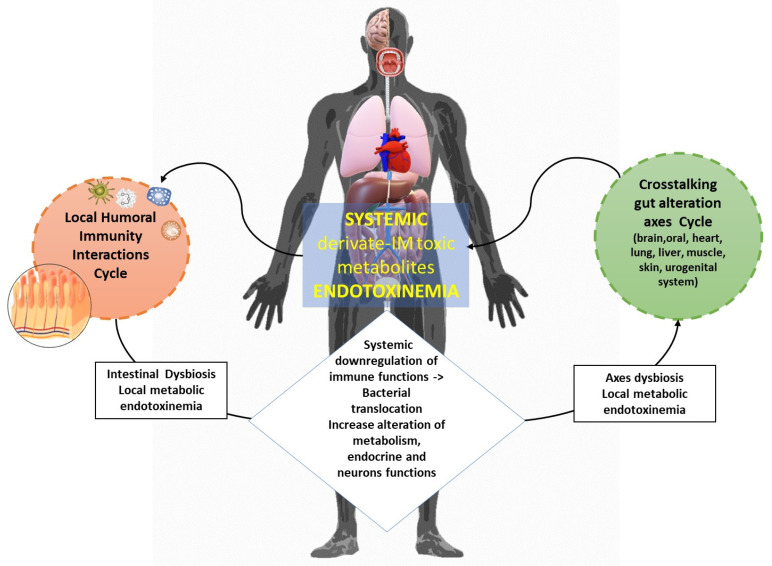
The hypothesis regarding the mechanisms involved in the occurrence of metabolic systemic endotoxemia suggest that IM dysbiosis initiates local immune interactions, leading to intestinal endotoxemia, which triggers a cascade of local and systemic reactions. This IM dysbiosis can affect some or all communication axes of the IM, resulting in generalized endotoxemia, further perpetuating the local and systemic imbalance. (Original figure by I.A. Charitos).

**Figure 10 ijms-25-02841-f010:**
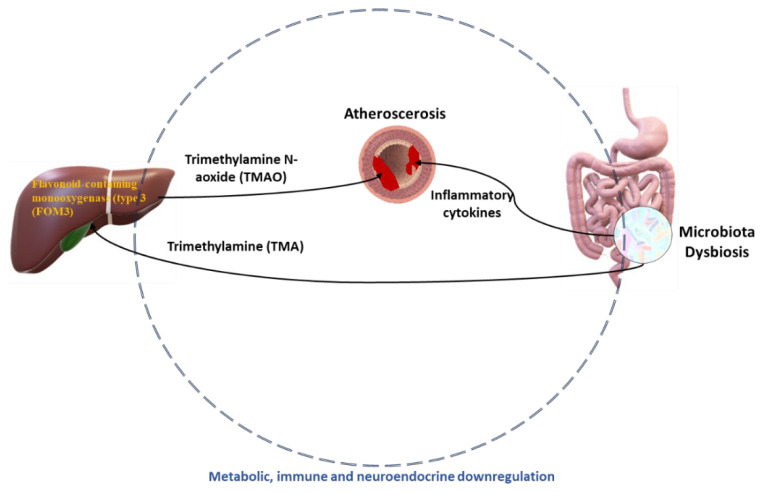
Initially, dysbiosis increases the population of some TMA-producing bacteria (such as those of *Prevotella* spp.), which contain flavonoids type 3 (FOM3). Subsequently, TMAO occurs in the liver, facilitating the formation of atheromasia plaques, resulting in local alterations that lead to endothelial sclerosis and damage to its functions. Furthermore, this inflammation and other reaction of the vessels will lead to a greater downregulation of immune, endocrine, and immune cycle homeostasis, worsening the state of cardiovascular alterations. (Original figure by I.A. Charitos).

**Figure 11 ijms-25-02841-f011:**
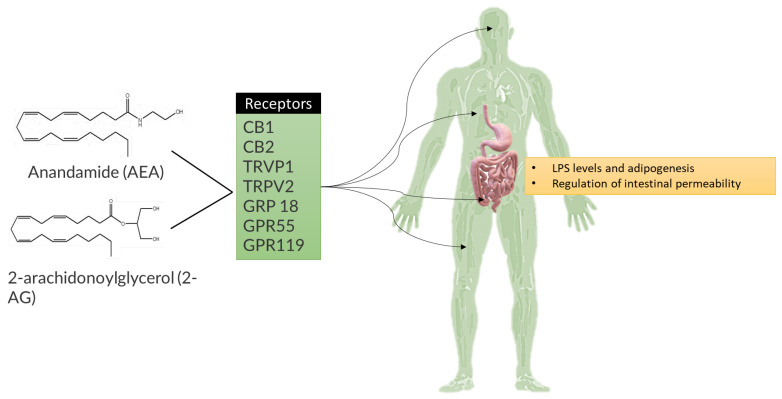
The figure illustrates the two endocannabinoids along with their receptors distributed throughout the body. Each of these endocannabinoids is found in varying quantities in the gastrointestinal organs, lungs, peripheral and central nerve system, bone marrow, and muscles. Numerous investigations utilizing specific antagonists and agonists have revealed that the endocannabinoid system governs not only intestinal permeability, but also plasma lipopolysaccharide (LPS) levels and adipogenesis. Endocannabinoids have been observed to boost occludin-1 protein mRNA expression, indicating a possible involvement in regulating intestinal permeability. (Original figure by I.A. Charitos).

**Table 1 ijms-25-02841-t001:** Compared constant versus intermitted excessive rates, significant changes can be observed in the metabolome in terms of carbohydrates, lipids, amino acids, nucleotides, and vitamins.

BPCO and Metabolome Changes during Work Rate Exercises	
Constant	Intermittent
**Increase:** alpha-ketoglutaric, malic, 2-hydroxybutyric and 3-hydroxybutyric acids **Decrease:** fructose-6-phosphate, 3-phosphoglyceric acid, l-carnitine and acylcarnitines	**Increase:** alpha-ketoglutaric, malic, 2-hydroxybutyric and 3-hydroxybutyric, citric acid, isocitric acid, lactic acids, inosine-5′-diphosphate, uric acid, ascorbic acid, pantothenic acid **Decrease:** fructose-6-phosphate, 3-phosphoglyceric acid, l-carnitine and acylcarnitines pyruvic, oxalic acids, in 5-hydroxymethyluridine, threonic acid, dehydroascorbic acid

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
