# Peer review of "Biomolecular Actions by Intestinal Endotoxemia in Metabolic Syndrome"

_ijms, 2024, doi:10.3390/ijms25052841_

Round 1

Reviewer 1 Report

Comments and Suggestions for Authors

The authors explore the intricate dynamics of Metabolic Syndrome (MetS), a cluster of metabolic disorders with far-reaching implications such as atherosclerosis and diabetes mellitus. Identifying six pivotal conditions within MetS, including visceral obesity, dyslipidemia, high blood pressure, insulin resistance, impaired glucose tolerance, and inflammation, the article underscores the growing role of metabolomics in scientific inquiry. In essence, the review article advocates for a comprehensive approach to understanding and addressing the complex interplay between MetS, the microbiome, and the metabolome, offering insights into preventive measures with implications for both MetS and associated respiratory conditions. The manuscript is well-written and adequately demonstrated. Nevertheless, there are areas that could be enhanced to uphold the quality of the research article.

`

1.     In the Introduction section, I suggest that the author also examine into the significance of Intestinal Endotoxemia in MetS.

2.     Maintain a consistent style when referring to Metabolic Syndrome (MetS) throughout the manuscript, as exemplified on Page 1, line 39.

3.     Italicize the names of isolates consistently across the entire manuscript, as demonstrated on Page 18, line 704.

4.     Place citations at the end of sentences; there are instances where sentences commence with citations, such as citation numbers 175, 143, 131, etc.

5.     I recommend that the authors enhance the clarity of the manuscript by providing an in-depth explanation of Biomolecular Interactions in MetS Progression, including insights into synergies, feedback loops, and molecular targets for intervention.

Reviewer 2 Report

Comments and Suggestions for Authors

The article is clearly and comprehensively written and is easy to read. Although it seems too long and at the same time the topic is not exhaustive in some respects

1. the summary lacks a summary and conclusion of the literature analysis. There is also no indication of the status of the manuscript.

2. introduction missing several citations in important statements

  line 63 DOI: 10.1111/jgh.15899; DOI: 10.3390/nu13061749

3. too little attention was paid to changes in glycemia and type II diabetes in terms of dysbiosis. adding paragraphs would be required

 DOI: 10.1080/10408398.2020.1809991; https://doi.org/10.3390/ijms21239212

4. line 165. it seems that determining metabolites rather than species diversity is a more useful parameter. The authors also describe the advantages of metabolomics over genomics, transcriptomics and proteomics, which they should point out as a direction for further research.

5. according to the current state of knowledge, the impact of the method of giving birth and antibiotic therapy on the microbiota and its metabolites applies to approximately 3 months and is not permanently related to

DOI: 10.3390/nu13041244

6. captions (names of figures) under figures should not contain citations. they are too long

7. if the title concerns MetS, units that are not directly related should be removed, line 407-414

8. The entire chapter 4 and 5 should be shortened and information previously mentioned in the article should not be repeated

9. the conclusions are too general, do not indicate the direction of further research and do not answer the possibility of using the described data in the MetS intervention (metabolic)

Reviewer 3 Report

Comments and Suggestions for Authors

Round 2

Reviewer 3 Report

Comments and Suggestions for Authors

The authors have addressed all my concerns.